# Factors Influencing Concordance of PD-L1 Expression between Biopsies and Cytological Specimens in Non-Small Cell Lung Cancer

**DOI:** 10.3390/diagnostics11101927

**Published:** 2021-10-18

**Authors:** Mohammed S. I. Mansour, Kim Hejny, Felicia Johansson, Joudy Mufti, Ante Vidis, Ulrich Mager, Annika Dejmek, Tomas Seidal, Hans Brunnström

**Affiliations:** 1Department of Pathology and Cytology, Halland Hospital Halmstad, SE-301 85 Halmstad, Sweden; kim.hejny@regionhalland.se (K.H.); felicia.a.johansson@regionhalland.se (F.J.); joudy.mufti@regionhalland.se (J.M.); ante.vidis@regionhalland.se (A.V.); tomas.seidal@regionhalland.se (T.S.); 2Division of Pathology, Department of Clinical Sciences Lund, Lund University, SE-221 00 Lund, Sweden; hans.brunnstrom@med.lu.se; 3Institute of Biomedicine, Sahlgrenska Academy, Gothenburg University, SE-405 30 Gothenburg, Sweden; 4Faculty of Natural Science, Kristianstad University, SE-291 88 Kristianstad, Sweden; 5Division of Respiratory and Internal Medicine, Department of Clinical Medicine, Halland Hospital Halmstad, SE-301 85 Halmstad, Sweden; ulrich.mager@regionhalland.se; 6Department of Translational Medicine in Malmö, Lund University, SE-205 02 Malmö, Sweden; annika.dejmek@med.lu.se; 7Department of Genetics and Pathology, Laboratory Medicine Region Skåne, SE-221 85 Lund, Sweden

**Keywords:** 28-8, pleural effusion, bronchial brush, bronchoalveolar lavage, cell block, biopsy, immunohistochemistry

## Abstract

PD-L1 expression assessed by immunohistochemical staining is used for the selection of immunotherapy in non-small cell lung cancer (NSCLC). Appropriate validation of PD-L1 expression in cytology specimens is important as cytology is often the only diagnostic material in NSCLC. In a previous study comprising two different cohorts of paired biopsies and cytological specimens, we found a fairly good cyto-histological correlation of PD-L1 expression in one, whereas only a moderate correlation was found in the other cohort. Therefore, that cohort with additional new cases was now further investigated for the impact of preanalytical factors on PD-L1 concordance in paired biopsies and cytological specimens. A total of 100 formalin-fixed paraffin-embedded cell blocks from 19 pleural effusions (PE), 17 bronchial brushes (BB), and 64 bronchoalveolar lavage (BAL) and concurrent matched biopsies from 80 bronchial biopsies and 20 transthoracic core biopsies from NSCLC patients were stained using the PD-L1 28-8 assay. Using the cutoffs ≥1%, ≥5%, ≥10%, and ≥50% positive tumour cells, the overall agreement between histology and cytology was 77–85% (*κ* 0.51–0.70) depending on the applied cutoff value. The concordance was better for BALs (*κ* 0.53–0.81) and BBs (*κ* 0.55–0.85) than for PEs (*κ* −0.16–0.48), while no difference was seen for different types of biopsies or histological tumour type. A high number of tumour cells (>500) in biopsies was associated with better concordance at the ≥50% cutoff. In conclusion, the study results suggest that PEs may be less suitable for evaluation of PD-L1 due to limited cyto-histological concordance, while a high amount of tumour cells in biopsies may be favourable when regarding cyto-histological PD-L1 concordance.

## 1. Introduction

Immune checkpoint inhibitors (ICIs) are potent anticancer drugs used in a variety of malignancies including non-small cell lung cancer (NSCLC) [1,2,3,4]. Immunohistochemical (IHC) staining for programmed death-ligand 1 (PD-L1) on formalin-fixed paraffin-embedded (FFPE) tissue specimens has been shown to predict clinical response to programmed death-1 (PD-1)/PD-L1 immune checkpoint therapies in NSCLC [3,5,6,7,8,9,10,11,12]; hence, PD-L1 expression in FFPE tissue is used for selecting patients for immunotherapy.

A significant part of NSCLC is diagnosed on cytological specimens [13,14,15]. The suitability of cytology for PD-L1 testing has been discussed [16,17,18], and the lack of comprehensive data and validation has prevented wide acceptance. PD-L1 expression in paired histological and cytological materials from NSCLC patients has been compared in several studies [19,20]. The overall agreement of PD-L1 reactivity has been reported to be relatively good between histology and cytology but with substantial variation between studies, both for concordance and proportion of positive PD-L1 NSCLC in cytological specimens.

Both methodological and biological aspects should be taken into account when comparing histology and cytology. The choice of PD-L1 assay, platform, and cut-off for a positive staining may have an impact on PD-L1 results regardless of specimen type, while fixation and preparation, specimen cell content, sampling site, heterogeneous expression, and interobserver agreement may be different for histological and cytological specimens [16,17,18]. Additionally, there is a mixture of both histological (biopsies, resections, and tissue microarrays) and cytological specimens (fine-needle aspirations, bronchoalveolar lavage [BAL], bronchial brush [BB], and pleural effusions [PE], etc.), which may be of importance as well.

In our previous study, we compared PD-L1 expression between histology and cytology in two different cohorts in southern Sweden [20]. We found a slightly lower rate of concordance in one of the cohorts, and in the present study, we further analysed that cohort regarding the impact of preanalytical factors on the concordance between biopsies and cytological specimens. We added new cases, creating a single cohort with a total of 100 cell blocks and paired biopsies from NSCLC specimens. PD-L1 reactivity was scored using four different cut-off levels.

The aim was to investigate the potential impact of different types of cytological specimens, different types of biopsies, histological tumour type, and specimen cell content on the concordance of PD-L1 expression between cytology and biopsies with NSCLC. To the best of our knowledge, no previous study has compared PD-L1 concordance between histology or subgroups of cytological samples, as previous studies included either one cytological sample type or a mixture of different sample types.

## 2. Materials and Methods

### 2.1. Specimen Collection

This retrospective study included 100 paired FFPE histological and cytological specimens with lung adenocarcinoma (66 paired cases) and squamous cell carcinoma (34 paired cases). The biopsies were bronchial biopsies in 80 cases and transthoracic core biopsies in 20 cases. The cytological specimens were PE in 19 cases, BB in 17 cases, and BAL in 64 cases.

The specimens had been submitted to the Department of Pathology and Cytology, Halland Hospital, Halmstad, Sweden, collected from May 2003 through March 2021 with staining for PD-L1 in 2016–2021 (30 of the 100 paired cases were at least 3 years old when stained for PD-L1). PD-L1 staining with the 28-8 assay (see below) had either been performed in the clinical setting or the specimens identically stained as part of the present study. For 83 (17 PE, 13 BB, and 53 BAL) of the cases, the paired specimens were included in a previous study [20].

All available paired samples that met the following criteria were included in this study: (i) the histological specimens were bronchial/lung biopsies, (ii) the cytological specimens were cell pellets from either PE, BB or BAL materials, (iii) the biopsy was obtained at the same time as the collection of the corresponding cytological specimen or within 4 weeks, (iiii) at least 100 tumour cells were present in each sample. The diagnoses of all included specimens were re-evaluated and verified on both histological and cytological materials by morphology and IHC/immunocytochemical (ICC) staining, respectively.

One cell block and one biopsy were stained with PD-L1 for each patient. For 95 paired cases, both the biopsy and cytological specimens were collected before the patient had received any oncological treatment; for 3 paired cases, both specimens were collected after the patient had received oncological treatment, while in 2 cases, one of the paired specimens was obtained before and the other after the patient had received oncological treatment.

### 2.2. Preparation Procedure and PD-L1 Staining

Preparation and staining of specimens were consistent over time and has previously been described [20]. In brief, the cell blocks were routinely available together with smears and ThinPrep^®^ slides in the clinical setting. The bronchial cytological specimens were put in CytoLyt^®^ for at least one hour, and after centrifugation, the cell pellets were manually transferred to the cassettes, which were fixed in neutral buffered formalin within 24 h. For PEs, cell pellets were instead transferred to the cassettes and fixed directly in formalin. Only bloody PEs were rapidly washed in CytoLyt^®^ up to three times just before formalin fixation.

The paired specimens were analysed for detection of PD-L1 using the PD-L1 IHC 28-8 pharmDx kit (Agilent/pharmDx, Santa Clara, CA, USA). Control slides produced from two cell lines, NCIH226 (PD-L1–positive) and MCF-7 (PD-L1–negative), provided by the manufacturer were included for each PD-L1 run. Additionally, an in-house control made from FFPE tissues from small intestine/appendix, tonsil, spleen, pancreas, and placenta was used on each sample slide.

Immunostaining was performed using an Autostainer Link 48, AS48430 (Agilent/Dako, Santa Clara, CA, USA) at the Department of Clinical Pathology and Cytology in Halmstad, in accordance with the manufacturer’s instructions.

### 2.3. Quantification of PD-L1 Expression and Estimation of Tumour Cell Proportion

PD-L1 reactivity was assessed using a conventional light microscope. The percentage of tumour cells expressing PD-L1 on the entire slide was semiquantified, and cases with the presence of a minimum of 100 viable tumour cells were considered adequate for evaluation. PD-L1 was scored <1%, ≥1%, ≥5%, ≥10%, and ≥50%, based on any intensity of complete or partial linear membranous staining of tumour cells according to the criteria recommended in the assessment manual (Agilent/pharmDx, Santa Clara, CA, USA) [21]. The number of tumour cells for the PD-L1 assessments was categorized using a four-tier scale: 100–300, >300–500, >500–1000, and >1000 tumour cells. Slides stained with haematoxylin-eosin and additional immunostains were available during the PD-L1 assessment. The same criteria of evaluation were applied to both histologic and cytologic specimens.

The PD-L1 scoring was performed blindly, independently, and without side-by-side comparison, first by a certified cytotechnologist (M.S.I.M.) and then by one or, when needed, two experienced cytopathologists (T.S.) and (H.B.) Cases with discordant scores between biopsy and cytology were reassessed by the involved investigators and for some cases also by a third cytopathologist (A.D.) Possible reasons for any discrepancy were discussed.

### 2.4. Statistical Analysis of Data

The association of PD-L1 expression with patient characteristics was evaluated using the Mann–Whitney U-test for age, Fisher’s exact test for sex, histologic tumour type, and type of biopsy, and Pearson’s Chi-square test for type of cytological specimen and specimen cell content.

The agreement between histology and cytology was assessed using Cohen’s Kappa (*κ*) at the ≥1%, ≥5%, ≥10%, and ≥50% cut-off levels (with bootstrapped 95% CI) [22]. According to the terminology of Altman [22], *κ* values were considered as poor (≤0.2.), fair (0.21–0.40), moderate (0.41–0.60), good (0.61–0.80), or very good (0.81–1.00). Overall percentage agreement (OPA), positive percentage agreement (PPA), and negative percentage agreement (NPA) with histology as the non-reference standard, and McNemar’s test were also calculated.

The relationship between PD-L1 discordance and preanalytical factors was analysed using Fisher’s exact test for histological tumour type and type of biopsy, and Chi-square test for the type of cytological specimen and specimen cell content.

All *p*-values were determined using two-sided tests and *p*-values > 0.05 were not considered statistically significant. Confidence intervals (CIs) using the modified Wald Method with Wilson score 95% were calculated according to website https://www.graphpad.com/quickcalcs/ (accessed on 27 August 2021). All other data analyses and summary graphs were produced using SPSS Statistics for Windows, Version 27.0 (IBM, Armonk, NY, USA).

## 3. Results

### 3.1. Characteristics of the Specimens

A total of 100 patients with a diagnosis of NSCLC and paired histological and cytological specimens which fulfilled inclusion criteria were analysed for PD-L1 expression.

Sixty of the cases were male and 40 were female, and the median age at diagnosis was 70 years (range 36–87 years). The PD-L1 expression and specimen cell content in the histological and cytological specimens, including details for different histological tumour types and different type of specimens, are shown in Table 1.

### 3.2. Relationship between PD-L1 Expression and Patient and Sample Characteristics

Expression of PD-L1 was evaluated in relation to patients’ age, sex, histologic type of NSCLC, type of biopsy for histological material, type of cytological specimen for cytological material, and specimen cell content at four cut-off levels (≥1%, ≥5%, ≥10%, and ≥50%).

Adenocarcinomas tended to be positive for PD-L1 more often than squamous cell carcinomas, regardless of specimen type, but this was only significant at cut-off ≥1% for cytological specimens (Fisher’s exact test, *p* = 0.012). Furthermore, for histological specimens, samples with a higher tumour cell content were less often PD-L1 positive, but this was only significant for cut-off ≥5% (Pearson’s Chi-square test, *p* = 0.045). This trend was not seen in cytological specimens. PD-L1 expression did not significantly correlate with age, sex, or specimen type. For full data, see Appendix A.

### 3.3. Concordance of PD-L1 Expression between Paired Histological and Cytological Specimens

The concordance of PD-L1 expression between the 100 paired histological and cytological NSCLC specimens was evaluated in several ways, including *κ*, OPA, PPA, and NPA (with histology as standard for PPA and NPA) for the different PD-L1 cut-offs ≥1%, ≥5%, ≥10%, and ≥50%. The results are found in Table 2, which also includes separate calculations for the different cytological specimen types (PE, BB, and BAL). As seen, the OPA varied between 68% and 94% depending on cytological specimen type (with the lowest concordance seen for PE specimens) and the used cut-off for a positive PD-L1 staining, while *κ* varied between −0.16 and 0.85. McNemar’s test was not statistically significant at any cut-off level.

The correlation of PD-L1 concordance to histological tumour types, type of biopsy, and cell content for histological and cytological material, respectively, was analysed. As previously, different cytological specimen types and different cut-offs for PD-L1 (≥1%, ≥5%, ≥10%, and ≥50%) were evaluated separately. The results are found in Table 3. The concordance did not differ with respect to the histological NSCLC type, type of biopsy, or specimen cell content for cytological material at any cut-off. However, when only analysing BAL cases, there was a significant difference for PD-L1 concordance depending on the specimen cell content in the biopsy, with higher concordance in cases with higher cell content in the biopsy, at ≥5% cut-off for PD-L1 (Pearson’s Chi-square test, *p* = 0.045). Additionally, a significantly higher PD-L1 concordance was also seen in cases with higher cell content in the histological material at ≥50% cut-off for PD-L1, both for all cases (Pearson’s Chi-square test, *p* = 0.017) and for BAL (*p* = 0.016).

Images from two different cases illustrating different histological types of NSCLC, type of cytological specimen, specimen cell content, and proportions of positive cells are shown in Figure 1 and Figure 2. Detailed data for all cases are shown in Appendix A.

## 4. Discussion

Given their use in clinical trials [1,2,3,4,7], biopsies are the established specimen type for IHC staining for PD-L1 in NSCLC. However, cytology is the only available material in a significant proportion of NSCLC cases [13,14,15,23,24], and typically offers enough material for various ancillary techniques [25,26,27,28,29,30]. As cytological specimens are handled differently, both compared to biopsies and between pathology departments, there is a need to evaluate cytology for reliability of PD-L1. Studies investigating the predictive value of PD-L1 on cytology in an immunotherapy-treated population are rare [31], while today, there is quite substantial data on PD-L1 staining in paired histological and cytological cases [19,20].

In our previous study [20], we found quite good cyto-histological concordance for PD-L1 in NSCLC overall but with considerable variation between investigations, both regarding concordance and frequency of PD-L1 positivity in cytological specimens. In one of our cohorts with paired biopsies and cytological specimens, a moderate concordance was seen [20]. Therefore, in this incremental study, we present a follow-up investigation with additional cases and exclusion of infrequent sample types (e.g., endobronchial ultrasound-guided and transthoracic fine needle aspirations) to enable conclusive statistics, and with further details such as multiple cutoff values (≥1% vs. ≥5% vs. ≥10% vs. ≥50%) and tumour cell content (100–300 cells vs. >300–500 cells vs. >500–1000 cells vs. >1000 cells). We further investigated the potential impact of different types of cytological specimens (PE vs. BB vs. BAL), different types of biopsies (bronchial biopsy vs. transthoracic core biopsy), and histological tumour type (adenocarcinoma vs. squamous cell carcinoma) on cyto-histological concordance of PD-L1 expression in NSCLC.

The most important findings of the present study were a lower biopsy-cytology PD-L1 concordance for PE samples than for bronchial cytology and that higher cell content (>500 tumour cells) for biopsies might lead to higher concordance. Additionally, it is noteworthy that a higher concordance was seen for ≥1% positive tumour cells than for other cut-offs, and that we did not see a difference in concordance between BB and BAL samples as in our previous study (which included fewer cases whereof some with cell blocks from mixed cytological specimens) [20].

The lower PD-L1 concordance seen for our PE cytology, which is in contrast with some studies [32,33], may be attributed to one or more several factors. The PE samples represent metastases while all biopsies in our cohort were from the primary tumour. Studies have reported that the PD-L1 expression may differ between primary tumour and metastasis with either gain or loss of expression in the metastasis [34,35,36,37], but no significant difference between metastatic sites has been described [38]. Interestingly, one study with mixed histological and cytological specimens reported a clear predictive value of PD-L1 in distant metastases but not in lymph nodes [39].

Cells in PE samples may be subject to degeneration, which should lead to generally weak or false negative staining. However, we found no obvious difference in the frequency of PD-L1 positivity in PE samples compared to bronchial cytology, and slightly more PD-L1 positive cases have been reported for PE than for other samples [40]. Additionally, we found no obvious difference in tumour cell content between cytological sample types. In our cohort, PE specimens were fixed in formalin compared to first CytoLyt^®^ and then formalin for BB and BAL samples, but this is unlikely to be a contributing factor [18].

Another factor is potential difficulty in evaluating cytology, and the Blueprint 2 study reported a clearly higher interobserver variability for cytology than for histology [41], but a moderate to near perfect interobserver agreement for cytological samples has been reported in other investigations [42,43,44]. In our experience, interobserver discordance is not higher for PE than other cytological specimens, but it may be discussed if training in PD-L1 evaluation specifically on cytological cases is needed.

The studies by Grosu et al. [32] and Zou et al. [33] reported a rather high PD-L1 concordance between histological specimens and PE cell blocks, with weighted *κ* values of 0.76 and 0.77, respectively, for PD-L1 <1% vs. 1–49% vs. ≥50% (the corresponding weighted *κ* was 0.25 in our PE subgroup). One obvious difference compared to our study is the mixture of biopsies and resections as histological specimens in these studies. Zou et al. also included biopsies from distant (including pleural) metastases and endobronchial ultrasound-guided needle aspirations as histological specimens, and in that study, a high rate of PD-L1 positivity was seen in PE samples in comparison with histological specimens (and compared to Grosu et al. and our study). The use of the 22C3 and SP263 assays, respectively, and inclusion of some non-adenocarcinoma cases by Grosu et al. and Zou et al. is probably of limited importance for cytological-histological PD-L1 concordance.

Histology- and cytology-based studies have reported a prevalence of PD-L1 positivity (≥1%) of 22–86% and 23–100%, respectively [6,20,40,45,46,47,48]. In the present investigation, the prevalence of PD-L1 positivity of 56% in biopsies and 53% in cytology is in line with a large series of unselected patients (56–63%) [40,49]. The strengths of our study include that we used the same PD-L1 assay and platform and stained all paired cases at the same time. Additionally, we only included concurrent sample cases, and to minimise the problem with interobserver variability, the assessment of PD-L1 reactivity was performed by the same examiners independently and together for consensus. The main limitation is the number of included cases, although the cohort is large in comparison with most previous investigations on paired histology-cytology cases. Another limitation is the retrospective design, with the inclusion of some old archival cases (not recommended in the clinical setting [16]), but this was equal for the biopsies and cell blocks, and probably did not have an impact on the outcome.

Too few patients in our cohort were treated with immunotherapy to allow for evaluation of the predictive value of PD-L1. However, it is of interest that of the seven patients who received first line mono-immunotherapy (all with PD-L1 >50% in biopsy), the two with cyto-histologically discordant PD-L1 expression (both with PD-L1 <50% in BAL) had progressive and stable disease, respectively, while one of the PD-L1-concordant cases showed stable disease and four partial response.

In conclusion, cell block preparations are suitable for routine PD-L1 assessment, but cyto-histological concordance may differ depending on the type of cytological sample. Especially, PE specimens showed limited cyto-histological PD-L1 concordance, which may indicate that this sample type may be less suitable for evaluation of PD-L1. Our results also suggest that tumour cell content for biopsies may affect concordance as well as frequency of PD-L1 positivity. However, further investigation is warranted.

## Figures and Tables

**Figure 1 diagnostics-11-01927-f001:**
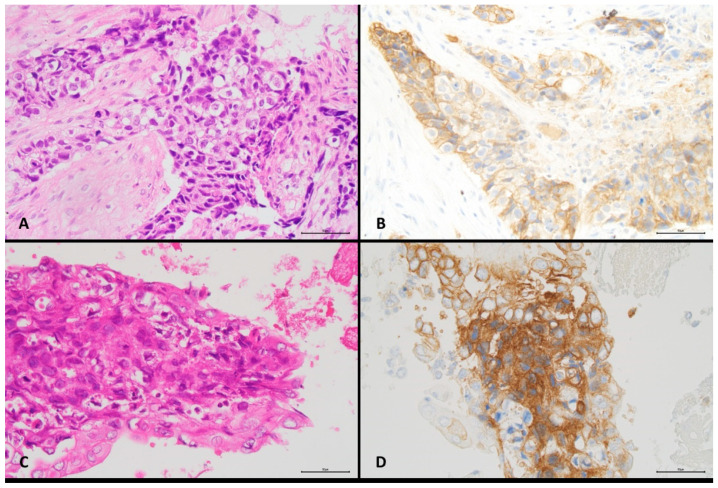
Concordant PD-L1 reactivity in paired histological and cytological specimens from a lung squamous cell carcinoma (original magnification ×400). (**A**) Bronchial biopsy, H&E staining, >500–1000 cells. (**B**) Bronchial biopsy, PD-L1 immunostaining, ≥50% positive malignant cells. (**C**) Bronchoalveolar lavage, H&E staining, >300–500 cells. (**D**) Bronchoalveolar lavage, PD-L1 immunostaining, ≥ 50% positive malignant cells.

**Figure 2 diagnostics-11-01927-f002:**
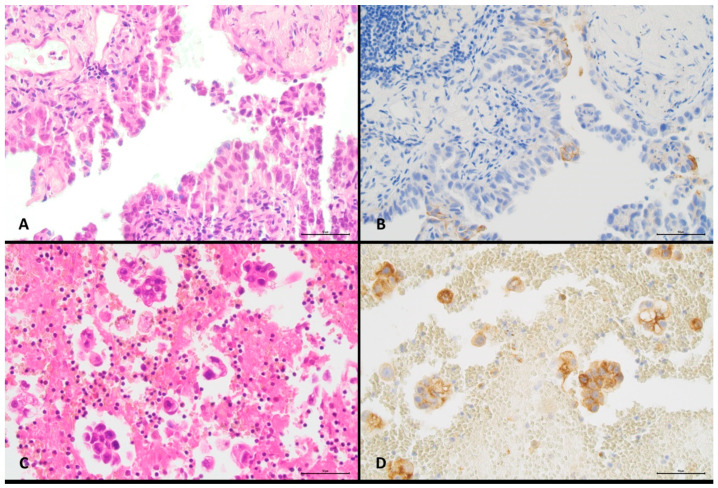
Discordant PD-L1 reactivity in paired histological and cytological specimens from a lung adenocarcinoma (original magnification ×400). (**A**) Transthoracic core biopsy, H&E staining, >500–1000 cells. (**B**) Transthoracic core biopsy, PD-L1 immunostaining, ≥5% positive malignant cells. (**C**) Pleural effusion, H&E staining, 100–300 cells. (**D**) Pleural effusion, PD-L1 immunostaining, ≥50% positive malignant cells.

**Table 1 diagnostics-11-01927-t001:** PD-L1 expression and cell content in paired histological and cytological non-small cell lung cancer specimens.

Characteristics	All Cases	PD-L1 Positive Tumour Cells					Cell Content Number of Tumour Cells			
		<1%	1–4%	5–9%	10–49%	≥50%	100–300	>300–500	>500–1000	>1000
Histological subtype, n (%)										
Adenocarcinoma	66/100									
Cytological samples		25/66 (38)	7/66 (11)	5/66 (8)	8/66 (12)	21/66 (32)	34/66 (52)	12/66 (18)	5/66 (8)	15/66 (23)
Biopsy samples		26/66 (39)	8/66 (12)	5/66 (8)	7/66 (11)	20/66 (30)	34/66 (52)	11/66 (17)	16/66 (24)	5/66 (8)
Squamous cell carcinoma	34/100									
Cytological samples		22/34 (65)	1/34 (3)	2/34 (6)	3/34 (9)	6/34 (18)	13/34 (38)	11/34 (32)	4/34 (12)	6/34 (18)
Biopsy samples		18/34 (53)	4/34 (12)	0/34 (0)	7/34 (21)	5/34 (15)	10/34 (29)	7/34 (21)	7/34 (21)	10/34 (29)
Type of cytological specimen, n (%)										
Pleural fluid	19/100	9/19 (47)	3/19 (16)	2/19 (11)	1/19 (5)	4/19 (21)	3/19 (16)	5/19 (26)	2/19 (11)	9/19 (47)
Bronchial brush	17/100	11/17 (65)	0/17 (0)	1/17 (6)	1/17 (6)	4/17 (24)	8/17 (47)	2/17 (12)	5/17 (29)	2/17 (12)
Bronchoalveolar lavages	64/100	27/64 (42)	5/64 (8)	4/64 (6)	9/64 (14)	19/64 (30)	36/64 (56)	16/64 (25)	2/64 (3)	10/64 (16)
Type of biopsy, n (%)										
Bronchial biopsy	80/100	36/80 (45)	10/80 (13)	4/80 (5)	9/80 (11)	21/80 (26)	37/80 (46)	15/80 (19)	18/80 (23)	10/80 (13)
Transthoracic core biopsy	20/100	8/20 (40)	2/20 (10)	1/20 (5)	5/20 (25)	4/20 (20)	7/20 (35)	3/20 (15)	5/20 (25)	5/20 (25)

**Table 2 diagnostics-11-01927-t002:** Agreement of PD-L1 expression in paired histological and cytological non-small cell lung cancer specimens with different cytological sample types specified.

PD-L1		All Cases	Pleural Effusions	Bronchial Brushes	Bronchoalveolar Lavages
Cutoff ≥1% positive cells (95% CI)	Kappa (κ)	0.70 (0.56–0.83)	0.48 (0.08–0.87)	0.55 (0.21–0.90)	0.81 (0.66–0.95)
	OPA	85 (77–91)	74 (51–89)	76 (52–91)	91 (81–96)
	PPA	89 (77–95)	70 (39–90)	100 (56–100)	92 (78–98)
	NPA	81 (67–90)	78 (44–95)	64 (35–85)	89 (71–97)
	McNemar	0.61	1.00	0.134	0.683
Cutoff ≥5% positive cells (95% CI)	Kappa (κ)	0.53 (0.37–0.70)	0.42 (−0.01–0.84)	0.65 (0.32–0.99)	0.53 (0.33–0.74)
	OPA	77 (68–84)	74 (51–89)	82 (58–95)	77 (65–85)
	PPA	73 (59–84)	57 (25–84)	100 (56–100)	72 (54–85)
	NPA	80 (67–89)	83 (54–97)	73 (43–91)	81 (64–91)
	McNemar	1.00	1.00	0.248	0.606
Cutoff ≥10% positive cells (95% CI)	Kappa (κ)	0.51 (0.34–0.69)	0.07 (−0.39–0.52)	0.64 (0.29–0.99)	0.56 (0.35–0.76)
	OPA	77 (68–84)	68 (46–85)	82 (58–95)	78 (66–87)
	PPA	71 (55–83)	20 (2–64)	100 (51–100)	75 (56–88)
	NPA	81 (69–89)	86 (59–97)	75 (46–92)	81 (65–91)
	McNemar	1.00	0.683	0.248	0.789
Cutoff ≥50% positive cells (95% CI)	Kappa (κ)	0.53 (0.34–0.72)	−0.16 (−0.33–0.00)	0.85 (0.57–1.00)	0.58 (0.36–0.80)
	OPA	82 (73–89)	68 (46–85)	94 (71–100)	83 (72–90)
	PPA	63 (44–79)	00 (00–55)	100 (45–100)	68 (46–85)
	NPA	89 (80–95)	87 (61–98)	92 (65–100)	89 (76–96)
	McNemar	0.81	0.683	1.00	1.00

Kappa (κ) is presented as range −1–1. OPA, PPA, and NPA are presented as percent with histology as the non-reference standard. McNemar analyses are presented as *p*-values; CI: confidence interval; Kappa (κ): Cohen’s kappa coefficient; NPA: Negative Percentage Agreement; OPA: Overall Percentage Agreement; PPA: Positive Percentage Agreement.

**Table 3 diagnostics-11-01927-t003:** Association of PD-L1 concordance for paired histological and cytological non-small cell lung cancer specimens with tumour type, specimen type, and cell content.

PD-L1		All Cases			Pleural Effusions			Bronchial Brushes			Bronchoalveolar Lavages		
		Concordant	Discordant	*p* Values	Concordant	Discordant	*p* Values	Concordant	Discordant	*p* Values	Concordant	Discordant	*p* Values
Cut-off ≥1% positive cells	All cases, n (%)	85/100 (85)	15/100 (15)		14/19 (74)	5/19 (26)		13/17 (77)	4/17 (24)		58/64 (91)	6/64 (9)	
	Histological subtype, n (%)			0.77			n/a			0.60			0.40
	Adenocarcinoma	55/66 (83)	11/66 (17)		14/19 (74)	5/19 (26)		6/7 (86)	1/7 (14)		35/40 (88)	5/40 (13)	
	Squamous cell carcinoma	30/34 (88)	4/34 (12)		0/0 (00)	0/0 (00)		7/10 (70)	3/10 (30)		23/24 (96)	1/24 (4)	
	Type of biopsy, n (%)			0.17			0.63			1.00			0.51
	Bronchial biopsy	70/80 (88)	10/80 (13)		8/10 (80)	2/10 (20)		10/13 (77)	3/13 (23)		52/57 (91)	5/57 (9)	
	Transthoracic core biopsy	15/20 (75)	5/20 (25)		6/9 (67)	3/9 (33)		3/4 (75)	1/4 (25)		6/7 (86)	1/7 (14)	
	Specimen cell amount “Cell blocks”, n (%)			0.34			0.23			0.28			0.52
	100–300	37/47 (79)	10/47 (21)		1/3 (33)	2/3 (67)		5/8 (63)	3/8 (38)		31/36 (86)	5/36 (14)	
	>300–500	20/23 (87)	3/23 (13)		4/5 (80)	1/5 (20)		1/2 (50)	1/2 (50)		15/16 (94)	1/16 (6)	
	>500–1000	8/9 (89)	1/9 (11)		1/2 (50)	1/2 (50)		5/5 (100)	0/5 (00)		2/2 (100)	0/2 (00)	
	>1000	20/21 (95)	1/21 (5)		8/9 (89)	1/9 (11)		2/2 (100)	0/2 (00)		10/10 (100)	0/10 (00)	
	Specimen cell amount “Biopsies”, n (%)			0.46			0.05			0.21			0.97
	100–300	37/44 (84)	7/44 (16)		9/10 (90)	1/10 (10)		3/6 (50)	3/6 (50)		25/28 (89)	3/28 (11)	
	>300–500	16/18 (89)	2/18 (11)		1/2 (50)	1/2 (50)		4/4 (100)	0/4 (00)		11/12 (92)	1/12 (8)	
	>500–1000	21/23 (91)	2/23 (9)		4/5 (80)	1/5 (20)		3/3 (100)	0/3 (00)		14/15 (93)	1/15 (7)	
	>1000	11/15 (73)	4/15 (27)		0/2 (00)	2/2 (100)		3/4 (75)	1/4 (25)		8/9 (89)	1/9 (11)	
Cut-off ≥5% positive cells	All cases, n (%)	77/100 (77)	23/100 (23)		14/19 (74)	5/19 (26)		14/17 (82)	3/17 (18)		49/64 (77)	15/64 (23)	
	Histological subtype, n (%)			0.21			n/a			1.00			0.14
	Adenocarcinoma	48/66 (73)	18/66 (27)		14/19 (74)	5/15 (26)		6/7 (86)	1/7 (14)		28/40 (70)	12/40 (30)	
	Squamous cell carcinoma	29/34 (85)	5/34 (15)		0/0 (00)	0/0 (00)		8/10 (80)	2/10 (20)		21/24 (88)	3/24 (13)	
	Type of biopsy, n (%)			0.77			0.63			1.00			1.00
	Bronchial biopsy	62/80 (78)	18/80 (23)		8/10 (80)	2/10 (20)		11/13 (85)	2/13 (15)		43/57 (75)	14/57 (25)	
	Transthoracic core biopsy	15/20 (75)	5/20 (25)		6/9 (67)	3/9 (33)		3/4 (75)	1/4 (25)		6/7 (86)	1/7 (14)	
	Specimen cell amount “Cell blocks”, n (%)			0.35			0.48			0.36			0.43
	100–300	34/47 (72)	13/47 (28)		3/3 (100)	0/3 (00)		6/8 (75)	2/8 (25)		25/36 (69)	11/36 (31)	
	>300–500	18/23 (78)	5/23 (22)		3/5 (60)	2/5 (40)		1/2 (50)	1/2 (50)		14/16 (88)	2/16 (13)	
	>500–1000	9/9 (100)	0/9 (00)		2/2 (100)	0/2 (00)		5/5 (100)	0/5 (00)		2/2 (100)	0/2 (00)	
	>1000	16/21 (76)	5/21 (24)		6/9 (67)	3/9 (33)		2/2 (100)	0/2 (00)		8/10 (80)	2/10 (20)	
	Specimen cell amount “Biopsies”, n (%)			0.32			0.69			0.45			0.045
	100–300	35/44 (80)	9/44 (21)		8/10 (80)	2/10 (20)		4/6 (67)	2/6 (33)		23/28 (82)	5/28 (18)	
	>300–500	11/18 (61)	7/18 (39)		1/2 (50)	1/2 (50)		4/4 (100)	0/4 (00)		6/12 (50)	6/12 (50)	
	>500–1000	18/23 (78)	5/23 (22)		4/5 (80)	1/5 (20)		3/3 (100)	0/3 (00)		11/15 (73)	4/15 (27)	
	>1000	13/15 (87)	2/15 (13)		1/2 (50)	1/2 (50)		3/4 (75)	1/4 (25)		9/9 (100)	0/9 (00)	
Cut-off ≥10% positive cells	All cases, n (%)	77/100 (77)	23/100 (23)		13/19 (68)	6/19 (32)		14/17 (82)	3/17 (18)		50/64 (78)	14/64 (22)	
	Histological subtype, n (%)			0.21			n/a			1.00			0.22
	Adenocarcinoma	48/66 (73)	18/66 (27)		13/19 (68)	6/19 (32)		6/7 (86)	1/7 (14)		29/40 (73)	11/40 (28)	
	Squamous cell carcinoma	29/34 (85)	5/34 (15)		0/0 (00)	0/0 (00)		8/10 (80)	2/10 (20)		21/24 (88)	3/24 (13)	
	Type of biopsy, n (%)			0.77			1.00			1.00			1.00
	Bronchial biopsy	62/80 (78)	18/80 (23)		7/10 (70)	3/10 (30)		11/13 (85)	2/13 (15)		44/57 (77)	13/57 (23)	
	Transthoracic core biopsy	15/20 (75)	5/20 (25)		6/9 (67)	3/9 (33)		3/4 (75)	1/4 (25)		6/7 (86)	1/7 (14)	
	Specimen cell amount “Cell blocks”, n (%)			0.75			0.78			0.57			0.53
	100–300	35/47 (75)	12/47 (26)		2/3 (67)	1/3 (33)		7/8 (88)	1/8 (13)		26/36 (72)	10/36 (28)	
	>300–500	17/23 (74)	6/23 (26)		3/5 (60)	2/5 (40)		1/2 (50)	1/2 (50)		13/16 (81)	3/16 (19)	
	>500–1000	8/9 (89)	1/9 (11)		2/2 (100)	0/2 (00)		4/5 (80)	1/5 (20)		2/2 (100)	0/2 (00)	
	>1000	17/21 (81)	4/21 (19)		6/9 (67)	3/9 (33)		2/2 (100)	0/2 (00)		9/10 (90)	1/10 (10)	
	Specimen cell amount “Biopsies”, n (%)			0.18			0.70			0.82			0.15
	100–300	34/44 (77)	10/44 (23)		7/10 (70)	3/10 (30)		5/6 (83)	1/6 (17)		22/28 (79)	6/28 (21)	
	>300–500	11/18 (61)	7/18 (39)		1/2 (50)	1/2 (50)		3/4 (75)	1/4 (25)		7/12 (58)	5/12 (42)	
	>500–1000	18/23 (78)	5/23 (22)		3/5 (60)	2/5 (40)		3/3 (100)	0/3 (00)		12/15 (80)	3/15 (20)	
	>1000	14/15 (93)	1/15 (7)		2/2 (100)	0/2 (00)		3/4 (75)	1/4 (25)		9/9 (100)	0/9 (00)	
Cut-off ≥50% positive cells	All cases, n (%)	82/100 (82)	18/100 (18)		13/19 (68)	6/19 (32)		16/17 (94)	1/17 (6)		53/64 (83)	11/64 (17)	
	Histological subtype, n (%)			0.11			n/a			0.41			0.51
	Adenocarcinoma	51/66 (77)	15/66 (23)		13/19 (68)	6/19 (32)		6/7 (86)	1/7 (14)		32/40 (80)	8/40 (20)	
	Squamous cell carcinoma	31/34 (91)	3/34 (9)		0/0 (00)	0/0 (00)		10/10 (100)	0/10 (00)		21/24 (88)	3/24 (13)	
	Type of biopsy, n (%)			0.75			1.00			1.00			1.00
	Bronchial biopsy	66/80 (83)	14/80 (18)		7/10 (70)	3/10 (30)		12/13 (92)	1/13 (8)		47/57 (83)	10/57 (18)	
	Transthoracic core biopsy	16/20 (80)	4/20 (20)		6/9 (67)	3/9 (33)		4/4 (100)	0/4 (00)		6/7 (86)	1/7 (14)	
	Specimen cell amount “Cell blocks”, n (%)			0.31			0.78			0.47			0.35
	100–300	41/47 (87)	6/47 (13)		2/3 (67)	1/3 (33)		8/8 (100)	0/8 (00)		31/36 (86)	5/36 (14)	
	>300–500	16/23 (70)	7/23 (30)		3/5 (60)	2/5 (40)		2/2 (100)	0/2 (00)		11/16 (69)	5/16 (31)	
	>500–1000	8/9 (89)	1/9 (11)		2/2 (100)	0/2 (00)		4/5 (80)	1/5 (20)		2/2 (100)	0/2 (00)	
	>1000	17/21 (81)	4/21 (19)		6/9 (67)	3/9 (33)		2/2 (100)	0/2 (00)		9/10 (90)	1/10 (10)	
	Specimen cell amount “Biopsies”, n (%)			0.017			0.70			0.33			0.016
	100–300	35/44 (80)	9/44 (21)		7/10 (70)	3/10 (30)		6/6 (100)	0/6 (00)		22/28 (79)	6/28 (21)	
	>300–500	11/18 (61)	7/18 (39)		1/2 (50)	1/2 (50)		3/4 (75)	1/4 (25)		7/12 (58)	5/12 (42)	
	>500–1000	21/23 (91)	2/23 (9)		3/5 (60)	2/5 (40)		3/3 (100)	0/3 (00)		15/15 (100)	0/15 (00)	
	>1000	15/15 (100)	0/15 (00)		2/2 (100)	0/2 (00)		4/4 (100)	0/4 (00)		9/9 (100)	0/9 (00)	

## Data Availability

The data presented in this study are available within the article and Appendix A.

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
