# Peer review of "Factors Influencing Concordance of PD-L1 Expression between Biopsies and Cytological Specimens in Non-Small Cell Lung Cancer"

_diagnostics, 2021, doi:10.3390/diagnostics11101927_

Round 1

Reviewer 1 Report

Thanks for giving me a chance to learn the association of PD-L1 expression between tissue and cytological samples in advanced non-small cell lung cancer. I read the manuscript and recommend you refer some questions and suggestions, below.

#1: I would like to ask you about the study period in Materials and Methods, saying all samples were collected from May 2003 through March 2021. As you know, it is important when the tissues and/or cytological specimen were obtained and stained for scoring PD-L1 TPS in NSCLC. Please explain it in detail in Materials and Methods, like your precedent article (Acta Cytol. 2021 Jul 7;1-9).

#2: Consequently, you mentioned “with staining for PD-L1 in 2016-2021”, however I think you stained very old specimens which were collected before 2016. If so, you should explain as one of limitation of this study in Discussion.

#3: I would like to confirm all specimens were stained by PD-L1 28-8 antibody in this study? Because both 22C3 and 28-8 antibodies were used in your precedent study.

#4: What do you think about the low kappa value in pleural effusion, demonstrating the large difference against another article (BMC Cancer. 2020 22;20:344). Yinying Zou et al. reported 0.774 of kappa value between pleural effusion and biopsy tissue in NSCLC.

#5: I afraid of the possibility of salami publication. What is most different and important message in this study? Final conclusion is similar between this study and your precedent study, I think.

After completely denying the possibility of salami publication, I agree the publication in the journal.

Thanks again

Author Response

Please, see attached file, Thanks

Reviewer 2 Report

The authors aimed to carry out a concordance study evaluating different specimen for PD-L1 detection and demonstrated that BAL, bronchial brush, and pleural effusion cell block are feasible. My comments are as follows:
1. The study used only one PD-L1 IHC antibody (Dako 28-8). Though the blueprint study showed a high concordance between Dako 28-8, 22C3, and Ventanna SP263, people may be still curious about the concordance of 22C3, SP263 and even SP142 in paired pathological specimens. Maybe testing other IHC antibody in a a small subset of study patients can be considered.

2.  Prior study showed EGFR mutant tumors tended to be low PD-L1 expressors.  Is the concordance in different specimens still good for EGFR mutant lung cancer ? 

3. Since PD-L1 is used to predict ICI efficacy, it is also important to correlate the expression level with clinical response if patients in the study cohort received immunotherapy. 

Author Response

Please, see attached file, Thanks
